# *Helicobacter pylori* infection is associated with fecal biomarkers of environmental enteric dysfunction but not with the nutritional status of children living in Bangladesh

Shah Mohammad Fahim[1]*, Subhasish Das[1], Md. Amran Gazi[1], Md. Ashraful Alam[1], Md. Mehedi Hasan[1], Md. Shabab Hossain[1], Mustafa Mahfuz[1,2], M Masudur Rahman[3], Rashidul Haque[4], Shafiqul Alam Sarker[1], Ramendra Nath Mazumder[1☯], Tahmeed Ahmed[1,5,6☯]

**1** Nutrition and Clinical Services Division, International Centre for Diarrheal Disease Research, Bangladesh (icddr,b), Dhaka, Bangladesh, **2** Faculty of Medicine and Life Sciences, University of Tampere, Finland, **3** Department of Gastroenterology, Sheikh Russel National Gastro Liver Institute & Hospital, Dhaka, Bangladesh, **4** Infectious Diseases Division, International Centre for Diarrheal Disease Research, Bangladesh (icddr,b), Dhaka, Bangladesh, **5** Department of Global Health, University of Washington, Seattle, Washington, United States of America, **6** James P. Grant School of Public Health, BRAC University, Dhaka, Bangladesh

☯ These authors contributed equally to this work.
* mohammad.fahim@icddrb.org

## Abstract

### Background

Because *Helicobacter pylori* (*H. pylori*) infection and Environmental Enteric Dysfunction (EED) follow a similar mode of transmission, there can be a complex interplay between *H. pylori* infection and EED, both of which can influence childhood growth. We sought to investigate the factors associated with *H. pylori* infection and identify its relationship with the fecal biomarkers of EED including Myeloperoxidase (MPO), Neopterin (NEO), Calprotectin, Reg1B and Alpha-1 antitrypsin (AAT), and nutritional status of the children.

### Methodology

Data from an on-going community-based nutrition intervention study was used for this analysis. Total 319 children aged between 12–18 months were evaluated at enrolment and at the end of a 90-day nutrition intervention. Multivariable linear regression with generalized estimating equations was done to examine the association of *H. pylori* infection with stool biomarker of EED and nutritional status of the children.

### Principal findings

One-fifth of the participants had *H. pylori* infection at both the time points, with 13.8% overall persistence. Children living in crowded households had higher odds of being infected by *H. pylori* (AOR = 2.02; 95% CI = 1.02, 4.10; p-value = 0.045). At enrolment, 60%, 99%, 69% and 85% of the stool samples were elevated compared to the reference values set for MPO,

**Data Availability Statement:** All relevant data are within the manuscript and its Supporting Information files.

**Funding:** This protocol is supported by the Bill and Melinda Gates Foundation under its Global Health Program. Project investment ID is OPP1136751. (https://www.gatesfoundation.org/How-We-Work/Quick-Links/GrantsDatabase/Grants/2015/11/OPP1136751). The funders had no role in the study design; collection, analysis, and interpretation of data; preparation, review, or approval of the manuscript; and decision to submit and publication of the manuscript.

**Competing interests:** The authors have declared that no competing interests exist.

NEO, AAT and Calprotectin in the non-tropical western countries. The proportions reduced to 52%, 99%, 67%, and 77% for the same biomarkers after the nutrition intervention. Infection with *H. pylori* had significant positive association with fecal AAT concentrations (Coefficient = 0.26; 95% CI = 0.02, 0.49; p-value = 0.03) and inverse relationship with Reg1B concentrations measured in the stool samples (Coefficient = -0.32; 95% CI = -0.59, -0.05; p-value = 0.02). However, *H. pylori* infection was not associated with the indicators of childhood growth.

## Conclusions

The study findings affirmed that the acquisition and persistence of *H. pylori* infection in the early years of life may exert an adverse impact on intestinal health, induce gut inflammation and result in increased intestinal permeability.

### Author summary

Infection with *H. pylori*, a substantial public health burden in the tropical countries, follows the similar mode of transmission analogous to Environmental Enteric Dysfunction (EED). There can be a complex interplay between *H. pylori* infection and EED–both of which can influence childhood growth–but the definite role of *H. pylori* infection contributing to EED and subsequent growth failure is poorly understood. In this study, the authors present data from an ongoing community-based nutrition intervention study and investigated the factors associated with *H. pylori* infection and identify its relationship with fecal biomarkers of EED and indicators of the nutritional status of the children hailing from a resource-poor urban settlement. They demonstrated the acquisition and persistence of *H. pylori* infection during early childhood. The study results also corroborate that infection with *H. pylori* had significant positive association with fecal Alpha-1 antitrypsin concentrations and an inverse relationship with Reg1B concentrations measured in stool samples of the children. The findings revealed in this study may contribute to a better understanding of the role of *H. pylori* infection in contributing to EED as well as alteration of gut function in the early years of life.

## Introduction

Infection with *Helicobacter pylori* (*H. pylori)* has emerged as a substantial public health burden over the past couple of decades [1]. The infection is highly prevalent in low-income countries and affects more than half of the global population [2]. The organism is obtained mostly by oral ingestion and induces chronic inflammation of the underlying gastric mucosa [3, 4]. The pathology is associated with diarrheal diseases, malnutrition and subsequent growth failure in children [5–8]. Evidence suggests that *H. pylori* infection is primarily acquired at the early years of life and can persist for a long period of time [9–11]. Infection acquired in the early age induces malabsorption and implicates in growth retardation [8]. The prevalence of the infection varies from around 10% to over 80% in children living in different regions of the world [12]. A birth cohort study, conducted in Bangladesh, showed that 50–60% of Bangladeshi children had the *H. pylori* infection by 2 years of their age [13]. Epidemiologic studies demonstrated that first two years of life is critical for growth [14] as well as for the acquisition of *H.*

*pylori* infection [15]. Consistent with those reports, many recent studies have exhibited the association of growth impairment with *H. pylori* infection, especially among those living in resource poor settings [8, 16–18]. Conversely, *H. pylori* has also been found to be linked with improved nutritional status in children, although the mechanism has not yet been elucidated [16, 19]. There is also evidence of having no relationship between infection with *H. pylori* and nutritional status of younger children [20]. The definite role of *H. pylori* on nutritional status of children is paradoxical and the findings are "mixed-bag". However, *H. pylori* infection can induce inflammatory responses as well as production of pro-inflammatory cytokines, and leads to malabsorption of essential nutrients [8]. Infection with *H. pylori* may also predisposes to multiple enteric pathogens resulting is altered intestinal health and function [8, 21].

Environmental Enteric Dysfunction (EED), an asymptomatic small intestinal pathology, has been implicated in linear growth failure of children less than two years of age [14, 22]. EED is characterized by persistent immune activation, gut inflammation and altered intestinal permeability resulting from chronic exposure to intestinal pathogens and frequent enteric infections [23–25]. The overall negative impact of EED on child growth and development, especially in their early years of life, is now well established [22, 26]. The condition has been described in the scientific literature since 1960s, but still there is no definite criteria to diagnose the ailment [22]. EED can be diagnosed through small intestinal biopsy which is considered to be the gold standard but difficult to perform in children owing to the invasiveness of the procedure [27]. However, several biomarkers have been tested as markers of EED and found to be associated with features of EED in previous studies [28–30]. Stool biomarkers including Myeloperoxidase (MPO), Neopterin (NEO), Calprotectin, Reg1B, and Alpha-1 antitrypsin (AAT) are the non-invasive alternatives proposed for the assessment of EED [27, 31]. MPO, Calprotectin, and NEO indicate intestinal inflammation, whereas AAT is a useful marker of enteric protein loss as well as intestinal permeability [32]. Reg1B is a newly proposed marker which points to epithelial tissue injury and subsequent repair in the small intestine [33]. EED is attributable to microbial contamination of food and water associated with poor sanitation and hygiene [34, 35]. Since *H. pylori* infection also follows the similar mode of transmission, there can be a complex interplay between the acquisition of *H. pylori* infection, EED and impaired growth in the first two years of life. Prior studies showed that *H. pylori* infection induces gastritis and results in protein losing enteropathy with evidence of resolution of the enteropathy by eradication of the infection [36, 37]. To that end, we hypothesized that *H. pylori* infections may contribute to and exacerbate EED and subsequent growth failure in children. But till date, no attempt was made to investigate the definite role of *H. pylori* infection contributing to EED and subsequent growth failure in children less than two years of age. Given the high prevalence of both *H. pylori* infection as well as EED in this patient population, investigation into how each condition influences the other and patient outcomes is of high importance. Therefore, we sought to investigate the factors associated with *H. pylori* infection and identify its relationship with fecal biomarkers of EED and indicators of the nutritional status of the children hailing from a resource-poor urban settlement in Dhaka, Bangladesh.

## Methods

### Ethics statement

The research protocol of this study (protocol no.: PR-16007) was approved by the Institutional Review Board of the International Center for Diarrheal Disease Research, Bangladesh (icddr, b), and written informed consent was obtained from the parents or legal guardians.

## Study design, site and population

Data from the Bangladesh Environmental Enteric Dysfunction (BEED) study was used to conduct this analysis. In brief, the BEED study is an ongoing community-based nutrition intervention study that is being conducted in the Mirpur area, a suburb located in the capital city of Bangladesh. In this study, children aged between 12 to 18 months either stunted [length-for-age z score (LAZ) <2] or at risk of stunting [LAZ = −1 to −2] are being enrolled for an intervention for 90 feeding days. The enrolled children receive an egg, 150 ml of whole milk, micronutrient sprinkles and nutritional counseling daily for 6 days in a week. A total of 319 children living in the slums of Mirpur area were included in this analysis. We included only those children who completed the nutrition intervention, and had data in both the time points–at enrollment and at the end of nutrition intervention. Exclusion criteria for enrollment in BEED study are: severe acute malnutrition, severe anemia, tuberculosis, presence of any congenital anomaly or deformity, suffering from diarrhoea or history of persistent diarrhoea in the preceding month, another family member already enrolled in the BEED study, and presence of any severe or chronic disease. The methodology of BEED study has been published previously [38].

## Data collection

Field staff collected the socio-economic and household information of the participants from the parents or caregivers at enrollment. Anthropometry was measured by the trained field staff following standard operating procedures (SOPs) based on the manuals of WHO and CDC [39, 40]. In order to ensure the consistency of an anthropometric measurement from one rater to another, we provided refresher's training to the field staff and estimated intra-class correlation coefficient (ICC) periodically every three months. Such training results in significant improvement of raters pertaining to anthropometric measurements at field site with a coefficient more than 0.9 for each of the scales. Indicators of nutritional status such as length-for-age z (LAZ), weight-for-age z (WAZ), and weight-for-height z (WHZ) scores were calculated using WHO anthropometry software. Blood and non-diarrheal stool samples were collected at baseline and after completion of 90-day nutrition intervention. Stool samples were obtained without using any fixative and frozen at −70˚C until analysis.

## Laboratory analysis

All the laboratory assays were carried out at icddr,b in Dhaka, Bangladesh. Blood samples were collected and centrifuged for 10 minutes at 4000 rotation per minute to separate the plasma. Aliquots were immediately stored at -80˚C till analysis. The inflammatory markers including high sensitivity CRP (Immundiagnostik, Bensheim, Germany) and AGP (Alpco, Salem, NH, USA) were analyzed from the plasma samples. Fecal biomarkers including AAT (Biovendor, Chandler, North Carolina), NEO (GenWay Biotech, San Diego, California), Reg1B (TechLab, Blacksburg, Virginia), Calprotectin (BÜHLMANN fCAL, Schönenbuch, Switzerland), and MPO (Alpco, Salem, New Hampshire) were measured in the stool samples using kits available for enzyme-linked immunosorbent assay (ELISA) following the instructions given by the manufacturers. Calibration curves were used to quantify the levels of each biomarker. In this study, fecal antigen test for *H. pylori* was employed to detect the *H. pylori* in the stool samples. This is a well-recognized non-invasive technique for the detection of *H. pylori* infection in the children [41]. Stool was analyzed for *H. pylori* antigen through ELISA using Amplified IDEIA™ Hp StAR™ (OXOID Limited, Hampshire, United Kingdom). Dual wavelength of $450/630_{nm}$ was used following the instruction of the manufacturer.

## Variables used in this analysis

We used *H. pylori* infection as the exposure variable. It was a binary categorical variable categorized based on the absorbance values derived from the stool ELISA results. Stool specimens with absorbance values ≥0.15 were considered positive and specimens with absorbance values <0.15 were considered negative for infection with *H. pylori*. Fecal biomarkers (e.g. MPO, NEO, Calprotectin, REG1B and AAT) and the indicators of nutritional status (e.g. LAZ, WAZ, and WHZ) were the outcome variables in our analyses. The covariates such as treatment of drinking water, source of drinking water, source of cooking water, hand washing practice after toilet, hand washing practice after helping the child to defecate, hand washing practice before cooking, separate space for kitchen, animal exposure at households, educational status of mothers and heads of households, and crowded living conditions were categorical variables. Crowded living condition was defined if more than 4 household members sleep in a single room[42]. Markers of systemic inflammation (e.g. CRP and AGP) were also included as covariates in this analysis (see the list of variables in S1 Table). We also divided the children enrolled in this study into four groups based on their infection with *H. pylori* and created a categorical variable–*H. pylor*i infection status. The categories of the variable are: a) children who had infection at enrollment but got cleared by the end of study, b) who acquired new infection during the study, c) children who remained infected at enrollment and at the end of nutrition intervention, and d) who remained non-infected in both the time points.

## Statistical analyses

Demographic and socio-economic characteristics were described by frequency with proportions for categorical variables, mean with standard deviation for symmetric continuous variables and median with inter-quartile ranges (IQR) for asymmetric continuous data. T-test, Wilcoxon rank-sum test and Pearson's chi-square test were applied to compare the baseline characteristics between the stunted and at risk of stunting children. The univariate Pearson's chi-square test was used to measure the differences in the prevalence of *H. pylori* infection both in stunted and at risk of stunting children at both the time points. We have identified the factors associated with *H. pylori* infection in non-diarrheal stool samples during enrollment using logistic regression model. Variables were assessed individually and were included in the multivariable logistic regression model if the p-value was found <0.2 in bivariate analysis. Education of household head was included in the model because of its previously reported association with the *H. pylori* infection in children [16, 42]. Additionally, the model was adjusted for age, sex, and nutritional status of the enrolled participants at enrolment.

Stool concentrations of all the fecal biomarkers (AAT, Reg1B, MPO, Calprotectin and NEO) were log-transformed. We then examined the association between *H. pylori* infection and stool biomarker concentrations and subsequently the association between *H. pylori* infection and indicators of nutritional status (LAZ, WAZ, and WLZ) of the children using multivariable linear regression with generalized estimating equations (GEE). In both the analyses, the family was Gaussian, identity was the link function and the correlation matrix was unstructured. The correlation matrix was selected based on the lowest quasi-likelihood under independence model criterion (QIC) value. Multicollinearity among the independent variables was checked for all the models using variance inflation factor (VIF) values. At first, bivariate analysis was done to explore the unadjusted effect of the variables on the outcomes using individual GEE model. Variables were included in the multivariable models if the p-value was found <0.2 in the bivariate analyses. In addition, all the estimates were adjusted for age and sex of the enrolled participants. We also performed multivariable linear regression analysis to test the association between *H. pylori* infection status and biomarker values at the end of nutrition

intervention. The biomarker values were log-transformed prior to analysis and the models were adjusted for age, sex, and nutritional status of the children at enrollment. Herein, we considered the children who remained non-infected as the reference group. A complete case analysis was applied for all the analyses and statistical significance was defined as a two-sided p-value<0.05. The statistical analyses were conducted using R version 3.5.1 (https://www.r-project.org, Foundation for Statistical Computing, Vienna, Austria) software.

## Results

A total of 319 children were included in this analysis. Among them 154 were stunted and 165 were at risk of being stunted children. The mean (±SD) age of the children was 14.5 (±2.1) months and 47.3% of the enrolled children were male. Almost 80% of the mothers received formal education. Water treatment rate was higher in the families of the children who are at risk of being stunted compared to the families of stunted children and it was found statistically significant (p = 0.047). The living condition of stunted children was more crowded than that of their peers. Compared to their counterparts, stunted children were more exposed to animals at the household level. The monthly family income of the stunted children was lower than that of at risk of stunting children. Table 1 describes the baseline characteristics of the enrolled children.

### Prevalence of *H. pylori* infection

The prevalence of infection with *H. pylori* at enrollment and at the end of nutrition intervention for both the stunted and at risk of being stunted children is presented in Fig 1. Although the proportion of *H. pylori* positivity was higher in the stool samples of the stunted children compared to at risk of stunting children at both the time points, the difference was not statistically significant (p-value>0.05). The prevalence was lower at enrollment compared to that of at the end of nutrition intervention, but it was not statistically different (p-value>0.05). The prevalence of persistent *H. pylori* infection as defined by the positivity of *H. pylori* infection at both the time points was 13.8% in this cohort of children (Fig 1). Persistence of *H. pylori* infection was more frequent in stunted children (15.6%) compared to at risk of being stunted children (12.1%). Here again, the difference was not found statistically significant (p-value>0.05).

### Distribution of fecal biomarkers in the stool samples

Overall, the fecal biomarker levels were much higher in the study participants compared to that of the standard in the non-tropical countries where the reference values for MPO, NEO, AAT and Calprotectin are <2,000 ng/mL, <70 nmol/L, <0.27 mg/g, <200 μg/g, respectively [43]. At enrolment, 60%, 99%, 69% and 85% of the stool samples were elevated compared to the reference values set for MPO, NEO, AAT and Calprotectin in the non-tropical western countries. The proportions reduced to 52%, 99%, 67%, and 77% for the same biomarkers after the nutrition intervention. In a recent study, the median values of fecal Reg1B concentration was found 30.8 and 16.5 μg/mL in the children of Bangladesh and Peru, respectively [44]. We have observed much higher concentrations of Reg1B compared to those findings in the stool samples of the children enrolled in this study. The median (IQR) concentration of Reg1B was 57.7 (31.1, 89.3) μg/mL at enrollment and it decreased to 48.3 (17.4, 82.6) μg/mL at the end of nutrition intervention. However, all the fecal biomarker values were reduced significantly after the 90-day nutrition intervention (p-value <0.05).

At both the time points, the concentrations of MPO, AAT and Calprotectin were higher in the stool samples of the *H. pylori* infected children. But only the difference in fecal AAT concentrations between infected and non-infected children at enrolment was found statistically significant (p-value = 0.04). Fecal NEO and Reg1B concentrations were lower in *H. pylori*

**Table 1. Descriptive characteristics of the stunted and at risk of being stunted children at enrollment.**

| Variables | Stunted (n = 154) | At risk of stunting (n = 165) | Total (N = 319) | p-value |
|---|---|---|---|---|
| **Socio-demographic variables** | | | | |
| Age in month, mean (SD) | 14.6 (2.1) | 14.4 (2.0) | 14.5 (2.1) | 0.39 |
| Gender (Male), n (%) | 88 (57.1%) | 63 (38.2%) | 151 (47.3%) | 0.001 |
| LAZ, mean (SD) | -2.9 (0.7) | -1.6 (0.3) | -2.2 (0.8) | <0.001 |
| WAZ, mean (SD) | -2.3 (0.8) | -1.4 (0.7) | -1.8 (0.9) | <0.001 |
| WHZ, mean (SD) | -1.1 (0.8) | -0.87 (0.9) | -0.98 (0.9) | 0.01 |
| Mothers received education, n (%) | 120 (77.9%) | 131 (79.4%) | 251 (78.7%) | 0.75 |
| Household head received education, n (%) | 98 (68.1%) | 117 (74.5%) | 215 (71.4%) | 0.22 |
| Water treatment, n (%) | 84 (54.5%) | 108 (65.5%) | 192 (60.2%) | 0.047 |
| Separate space for kitchen, n (%) | 124 (80.5%) | 145 (87.9%) | 269 (84.3%) | 0.07 |
| Always wash hand before cooking, n (%) | 14 (9.1%) | 24 (14.6%) | 38 (11.9%) | 0.13 |
| Always wash hand after toilet, n (%) | 103 (66.9%) | 126 (76.4%) | 229 (71.8%) | 0.06 |
| Always wash hand after child defecation, n (%) | 85 (55.2%) | 99 (60%) | 184 (57.7%) | 0.39 |
| Improved toilet, n (%) | 99 (64.3%) | 108 (65.5%) | 207 (64.9%) | 0.83 |
| Crowded living conditions, n (%) | 43 (27.9%) | 31 (18.8%) | 74 (23.2%) | 0.05 |
| Animal exposure in household, n (%) | 15 (9.9%) | 8 (4.9%) | 23 (7.3%) | 0.09 |
| Monthly family income (USD)*, mean (SD) | 167.3 (84.1) | 190.6 (117.3) | 179.4 (103.1) | 0.04 |
| **Markers of systemic inflammation** | | | | |
| CRP (mg/l), median (IQR) | 1.2 (0.4, 3.2) | 1.0 (0.6, 4.1) | 1.1 (0.5, 3.6) | 0.69 |
| AGP (mg/dl), median (IQR) | 96.3 (70.8, 127.6) | 85.1 (64.1, 122.5) | 92.6 (66.5, 125.6) | 0.11 |
| **Fecal biomarkers of EED** | | | | |
| MPO (ng/mL), median (IQR) | 2740.5 (1457.2, 5505.2) | 2266.0 (1427.0, 4758.0) | 2438.0 (1434.0, 5398.0) | 0.27 |
| NEO (nmol/L), median (IQR) | 2902.0 (1907.0, 4269.0) | 3226.0 (1732.0, 5149.0) | 3068.0 (1850.0, 4548.0) | 0.26 |
| AAT (mg/g), median (IQR) | 0.46 (0.23, 0.68) | 0.47 (0.25, 0.64) | 0.46 (0.24, 0.67) | 0.97 |
| Calprotectin (μg/g), median (IQR) | 524.2 (266.2, 1025.5) | 672.4 (327.9, 1074.8) | 598.3 (300.1, 1041.5) | 0.09 |
| Reg1B (μg/mL), median (IQR) | 62.2 (33.4, 91.7) | 51.9 (30.8, 85.1) | 57.7 (31.4, 89.3) | 0.31 |

*1 USD = 84.21 BDT was used as conversion rate

infected children, which was found statistically insignificant at enrollment but significant only for Reg1B at the end of nutrition intervention (p-value = 0.01). There was no statistically significant difference in the fecal biomarker concentrations of children with persistent infection compared to those without persistent infection (p-value>0.05).

## Factors associated with *H. pylori* infection

Multivariable logistic regression model demonstrated that children living in crowded households had higher odds of being infected by *H. pylori* (AOR = 2.02; 95% CI = 1.02, 4.10; p-value = 0.045) in this cohort after controlling for the age, sex, nutritional status at enrollment, mother's education, education received by household head, and water treatment. No other socio-demographic factor demonstrated any statistically significant association with the *H. Pylori* infection (Table 2).

## Association of *H. pylori* infection with the fecal biomarkers of EED

Table 3 showed the association of *H. pylori* infection with the fecal biomarkers of EED. No significant association was observed between the infection and fecal levels of MPO, NEO, and

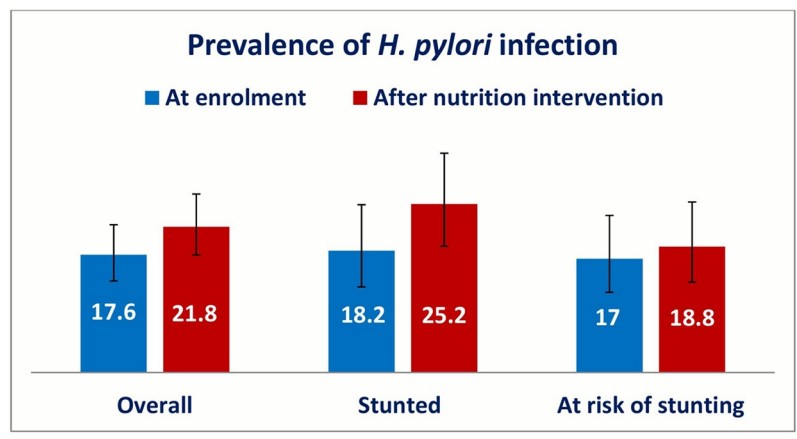

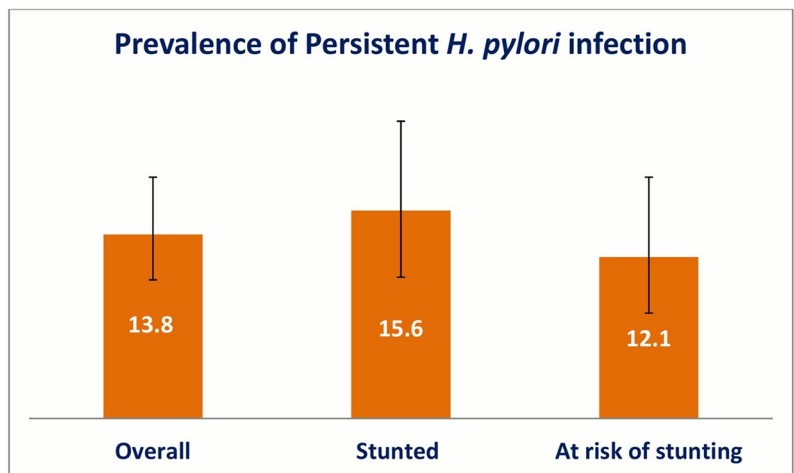

**Fig 1. Prevalence of *Helicobacter pylori* infection in children living in Bangladesh**

Calprotectin, both in bivariate and multivariable analysis using GEE. However, *H. pylori* infection was significantly associated with the fecal concentrations of AAT (Coefficient = 0.26; 95% CI = 0.02, 0.49; p-value = 0.03) after adjusting for age, sex, nutritional status at enrollment, mother's education, crowded living conditions, water treatment, hand washing practice of mother after toilet, CRP, and AGP. A statistically significant negative association was observed between *H. pylori* infection and fecal concentrations of REG1B (Coefficient = -0.32; 95% CI =

**Table 2. Factors associated with *Helicobacter pylori* infection in children during enrollment.**

| Variables | OR (95% CI) | p-value | AOR (95% CI) | p-value |
|---|---|---|---|---|
| Age | 1.11 (0.96, 1.27) | 0.16 | 1.08 (0.93, 1.26) | 0.29 |
| Sex (female) | 0.68 (0.38, 1.21) | 0.19 | 0.62 (0.33, 1.17) | 0.14 |
| Nutritional status (At risk of stunting) | 0.92 (0.52, 1.64) | 0.78 | 1.24 (0.66, 2.33) | 0.51 |
| Crowding (> 4 people sleep per room) | 1.94 (1.04, 3.64) | 0.04 | 2.02 (1.02, 4.10) | 0.045 |
| Mother received education | 0.55 (0.29, 1.05) | 0.07 | 0.62 (0.28, 1.37) | 0.24 |
| Household head received education | 0.71 (0.38, 1.34) | 0.29 | 0.99 (0.47, 2.13) | 0.99 |
| Treatment of water | 0.66 (0.37, 1.18) | 0.16 | 0.88 (0.46, 1.67) | 0.69 |

**Table 3. Association of *Helicobacter pylori* infection with the fecal biomarkers of EED using GEE[¶].**

| Variables | AAT, mg/g | MPO, ng/mL | NEO, nmol/L | Calprotectin, µg/g | Reg1B, µg/mL |
|---|---|---|---|---|---|
| Age in days | -0.004(-0.05, 0.04) | -0.02 (-0.07, 0.03) | -0.05 (-0.10, -0.001)* | -0.02 (-0.07, 0.04) | -0.08 (-0.13, -0.02)* |
| Sex (female) | -0.14 (-0.34, 0.06) | 0.12 (-0.09, 0.33) | 0.03 (-0.18, 0.24) | -0.02 (-0.20, 0.25) | 0.12 (-0.10, 0.35) |
| Nutritional status (At risk of stunting) | -0.07 (-0.27, 0.13) | -0.02 (-0.24, 0.19) | 0.06 (-0.15, 0.28) | -0.10 (-0.33, 0.12) | -0.03 (-0.25, 0.20) |
| Mothers received education | 0.19 (-0.06, 0.45) | 0.20 (-0.07, 0.47) | | | 0.29 (0.01, 0.56)* |
| Water treatment | 0.01 (-0.22, 0.23) | | | | |
| Separate space for kitchen | | | -0.24 (-0.56, 0.08) | | |
| Animal exposure in household | | | | -0.42 (-0.84, -0.002) | |
| Always wash hand after child defecation | | | -0.14 (-0.39, 0.11) | | -0.18 (-0.45, 0.09) |
| Always wash hand after toilet | 0.21 (-0.03, 0.45) | 0.26 (0.02, 0.49)* | -0.01 (-0.27, 0.29) | | -0.02 (-0.31, 0.28) |
| Crowding | -0.09 (-0.33, 0.15) | 0.15 (-0.10, 0.40) | | | |
| CRP | 0.01 (-0.01, 0.03) | 0.02 (-0.01, 0.04) | -0.03 (-0.05, -0.01)* | 0.01 (-0.01, 0.04) | |
| AGP | 0.001 (-0.001, 0.003) | 0.001 (-0.001, 0.003) | | 0.003 (0.0003, 0.005)* | |
| *Helicobacter pylori* infection | 0.26 (0.02, 0.49)* | 0.14 (-0.11, 0.39) | -0.02 (-0.26, 0.23) | -0.07 (-0.34, 0.20) | -0.32 (-0.59, -0.05)* |

[¶]Each column represents an individual model. The adjusted coefficient with 95% confidence interval (CI) has been reported.

The asterisk (*) denotes the statistical significance with a p-value < 0.05.

Abbreviations used: AAT, alpha-1 antitrypsin; MPO, myeloperoxidase; NEO, neopterin; CRP, C-reactive protein; AGP, Alpha-1-acid glycoprotein.

-0.59, -0.05; p-value = 0.02) after adjustment for age, sex, nutritional status at enrollment, mother's education, and hand washing practice of mother after defecating the child.

We observed statistically significant association between *H. pylori* infection status and biomarkers of EED at the end of nutrition intervention (Table 4). The multivariable linear regression analyses showed that children who were infected at enrolment had significantly higher fecal concentrations of AAT (p-value = 0.03), MPO (p-value = 0.01), and calprotectin (p-value = 0.02) at the end of nutrition intervention compared to the children who remained non-infected. Children who acquired infection during study (p-value = 0.03) and who remained infected in the both the time points (p-value = 0.006) had significantly lower concentrations of fecal Reg1B compared to the reference group. Fecal NEO concentration was significantly lower (p-value < 0.001) in children who acquired infection during the study in comparison to the children who had no infection at all.

**Table 4. Association of *Helicobacter pylori* infection status with the fecal biomarkers of EED using multivariable linear regression analysis[¶].**

| Variables | AAT, mg/g | MPO, ng/mL | NEO, nmol/L | Calprotectin, µg/g | Reg1B, µg/mL |
|---|---|---|---|---|---|
| Age in days | -0.03 (-0.09, 0.03) | -0.04 (-0.08, 0.01) | -0.08 (-0.14, -0.02) * | -0.05 (-0.11, 0.01) | -0.11 (-0.19, -0.03) * |
| Sex (female) | -0.11(-0.31, 0.09) | -0.003 (-0.20, 0.19) | 0.16 (-0.04, 0.36) | -0.12 (-0.36, 0.12) | 0.08 (-0.23, 0.39) |
| Nutritional status (At risk of stunting) | -0.19(-0.39, 0.01) | 0.05(-0.15, 0.25) | -0.08 (-0.28, 0.12) | 0.03 (-0.21, 0.27) | -0.06 (-0.39, 0.27) |
| *Helicobacter pylori* infection status (Ref: non-infected) | | | | | |
| Infected at enrolment | 0.58(0.05, 1.11) * | 0.67 (0.16, 1.18) * | 0.19 (-0.32, 0.70) | 0.73 (0.10, 1.36) * | 0.11 (-0.73, 0.95) |
| Infected acquired during study | 0.33(-0.04, 0.70) | 0.30(-0.07, 0.67) | -0.72 (-1.09, -0.35) * | 0.29 (-0.14, 0.72) | -0.68 (-1.27, -0.10) * |
| Remained infected | -0.09(-0.38, 0.20) | 0.14(-0.15, 0.43) | -0.09 (-0.38, 0.20) | 0.27 (-0.08, 0.62) | -0.67 (-1.14, -0.20) * |

[¶]Multivariable linear regression was applied considering the biomarker values at the end of nutrition intervention as the outcome variables. Each column represents an individual model. Biomarker values were log-transformed prior to analysis. Adjusted coefficient values with 95% CI have been reported in the table.

The asterisk (*) sign indicates the statistical significance.

## Association of *H. pylori* infection with nutritional status of the children

In multivariable analysis, infection with *H. pylori* was not associated with LAZ score of the children after adjusting for age, sex, maternal height, mother's education, education received by of household head, crowded living conditions, separate space for kitchen, water treatment, hand washing practice of mother after toilet, and monthly family income. *H. pylori* infection did not have any significant association with WAZ score of the children after controlling for the variables named age, sex, maternal height, mother's education, education received by of household head, crowded living conditions, separate space for kitchen, water treatment, improved toilet, hand washing practice of mother after toilet, hand washing practice of mother after defecating child, Animal exposure, AGP, and monthly family income. No statistically significant association was observed between the *H. pylori* infection and WLZ score of the children after adjustment for the above-mentioned confounding variables (Table 5).

## Discussion

Our study results revealed that children living in crowded households had higher odds of being infected by *H. pylori*. We observed nearly one-fifth of the participants had *H. pylori* infection at both the time points, with 13.8% overall persistence. Infection with *H. pylori* was positively associated with fecal AAT concentrations. An inverse association was observed between the infection and fecal Reg1B concentrations of these children. In addition, a positive association was reported between *H. pylori* infection and fecal concentrations of AAT, MPO, and calprotectin at the end of nutrition intervention in children who were infected at enrollment compared to the children who remained non-infected during the study period. On the other hand, fecal Reg1B concentration measured at the end of the study was lower in children who acquired infection during the study and who remained infected in both the time points.

**Table 5. Association of *Helicobacter pylori* infection with the indicators of nutritional status in children[¶].**

| Variables | LAZ | WAZ | WLZ |
|---|---|---|---|
| Age | -0.02 (-0.06, 0.2) | -0.02 (-0.07, 0.03) | -0.03 (-0.08, 0.02) |
| Sex (female) | 0.34 (0.17, 0.51) * | 0.33 (0.14, 0.53) * | 0.24 (0.04, 0.43) * |
| Mothers received education | 0.18 (-0.06, 0.41) | 0.26 (-0.02, 0.53) | 0.18 (-0.10, 0.46) |
| Household head received education | 0.19 (-0.03, 0.40) | 0.18 (-0.07, 0.43) | 0.16 (-0.09, 0.41) |
| Separate space for kitchen | 0.02 (-0.23, 0.28) | 0.14 (-0.18, 0.44) | 0.08 (-0.24, 0.40) |
| Water treatment | 0.001 (-0.20, 0.19) | 0.22 (0.001, 0.45) | 0.26 (0.04, 0.49) * |
| Improved toilet | | 0.08 (-0.13, 0.29) | 0.19 (-0.03, 0.40) |
| Always wash hand after toilet | 0.14 (-0.06, 0.34) | 0.02 (-0.25, 0.29) | |
| Always wash hand after child defecation | | -0.11 (-0.34, 0.13) | -0.14 (-0.35, 0.07) |
| Monthly family income | 5.2e-06 (-4.8e-06, 0.00001) | 6.2e-06 (-4.5e-06, 0.00002) | 5.7e-06 (-5.2e-06, 0.00002) |
| Crowded living condition | -0.07 (-0.28, 0.14) | -0.10 (-0.33, 0.14) | -0.07 (-0.30, 0.17) |
| Animal exposure in the households | | -0.40 (-0.76, -0.03) * | -0.32 (-0.70, 0.05) |
| Maternal height | 0.02 (0.01, 0.04) * | 0.02 (0.01, 0.04) * | 0.02 (-0.002, 0.04) |
| AGP | | -0.002 (-0.002, -0.001) * | -0.002 (-0.003, -0.001) * |
| *Helicobacter pylori* infection | 0.05 (-0.05, 0.15) | 0.08 (-0.04, 0.20) | 0.06 (-0.09, 0.21) |

[¶]Each row represents an individual model. Adjusted coefficient values with 95% confidence interval have been reported in the table.

The asterisk (*) sign indicates the statistical significance.

Past evidence has been limited to the association between *H. pylori* infection and childhood growth only, and no research was done to explore the role of *H. pylori* infection on the changes in EED biomarkers. To our knowledge this is the first attempt to investigate the relationship between *H. pylori* infection, fecal biomarkers of EED, and subsequent child growth in Bangladeshi children living in an urban community. Our findings, which were based on a well-designed community-based nutrition intervention study, provided an accurate estimate of the burden of *H. pylori* infection as well as its persistence in the children of an urban area in Bangladesh. Moreover, the results of the study reinforce the hypothesis that *H. pylori* infection may contribute to the exacerbation of altered gut health as well as EED in young children living in poor environment.

It is known that *H. pylori* can persist at a high rate in the gastrointestinal tract of people living in resource limited settings [11, 45, 46]. The infection is inversely associated with the living conditions as well as the practice of hygiene and sanitation [47]. Previous reports indicated that children from the households with greater number of inhabitants are more prone to have *H. pylori* infection [2, 48, 49]. Moreover, it is hypothesized that the infection with *H. pylori* transmits through oral-oral route and within families [50–52]. Our finding also supports the transmission within families and goes in line with the previous evidence of having association between *H. pylori* infection and excessive household members.

Approximately one-fifth of our study participants, irrespective of their nutritional status, were infected at enrollment. The prevalence increased at the end of nutrition intervention, although it was statistically insignificant. However, the prevalence rate that we have observed was lower compared to the prior researches conducted in this country [11, 52]. A birth cohort study conducted in rural Bangladesh reported the seroprevalence as 47.6% in children at the end of two years of age [53]. Another study that followed the children of an urban slum up to 2 years of age stated 50% of *H. pylori* positivity using the fecal antigen test in Bangladesh [54]. Earlier studies conducted in India, Argentina, Ethiopia, and Brazil have reported the prevalence of *H. pylori* infection as 22%, 25%, 48%, and 55%, respectively in children during their early years [9, 16, 54]. Perhaps, improvements in the living standard including access to safe water, improved sanitation, and better hygiene practice played a potential role in reducing the prevalence of the infection in this community [55]. Moreover, the study participants enrolled in this study are from 12 to 18 months of age. The younger age of the participants can be another possible explanation for such lower rate of positivity compared to earlier studies, because the frequency of *H. pylori* acquisition increases with the increase of age up to 10 years of age [53, 56]. However, since we observed the prevalence of *H. pylori* infection in children less than two years of age, it substantiates the evidence of the acquisition of *H. pylori* infection in early childhood, even before two years of age. Our results also confirm the persistence of *H. pylori* infection in the under-2 children enrolled in this study.

In this study, the fecal biomarker values were much elevated in this cohort of children compared to the reference values for those living in western countries. This finding is consistent with the previous documents published earlier using data from both urban and rural areas of Bangladesh [57, 58]. However, such elevation indicates the widespread perturbation of gut health and integrity in children living in resource limited urban settlement in Bangladesh.

We observed that infection with *H. pylori* was associated with the increased levels of AAT in the stool samples of the children. Additionally, fecal AAT measured at the end of nutrition intervention was higher in children who were infected at enrollment but recovered by the end of study. AAT is a protein that is released by neutrophils during infection or inflammatory conditions [59, 60]. Since AAT is not produced in the intestine, its presence in the gut lumen can be considered as a measure of gut inflammation [59]. AAT also reflects protein loss due to disruption of mucosal barrier and has been established as a marker of increased intestinal

permeability [58, 59]. Because AAT can be used for assessing both intestinal inflammation and increased gut permeability, two important domains of EED, several studies measured fecal AAT as a biomarker of EED [14, 22, 27, 61]. Prior report showed that fecal AAT as well as "EED composite score" comprising AAT were associated with impaired linear growth in children [22, 62]. To that end, AAT measured in the stool samples can be a better readout for diagnosing EED. However, there remains paucity of research investigating the relationship between *H. pylori* infection and fecal concentrations of AAT. An old case report of a 37 year old man with *H. pylori* infection documented an elevated level of AAT in his stool samples [63]. Another report showed that *H. pylori* infection in children was associated with acute gastritis and protein losing enteropathy [36, 64]. They observed resolution of enteropathy as well as improvement of the gastritis after clearance of the organism, and therefore, hypothesized the protein losing enteropathy as a consequence of gastritis caused by *H. pylori* [36]. Recent evidence has shown that *H. pylori* bacteria shed outer membrane vesicles (OMV) [65]. OMV is known to be responsible for inducing intestinal barrier dysfunction and tight junction disruption [66]. In accordance with above mentioned old reports as well as the recent evidence pertaining to OMV, our result also corroborates that acquisition of *H. pylori* infection may contribute to increased intestinal permeability and gut inflammation in infected children. This finding is in favor of our hypothesis and would contribute to advance further understanding on the complex interplay between infection with *H. pylori*, impaired intestinal health and EED.

Similar to AAT, children with *H. pylori* infection had higher concentrations of MPO and calprotectin at the end of nutrition intervention. Both MPO and calprotectin are released from neutrophils and indicates intestinal inflammation [32, 67, 68]. MPO has been tested as a biomarker of EED in several studies [14, 22, 27, 28]. Previously, MPO was found to be associated with markers of intestinal permeability including AAT [32, 57, 59]. A positive association of calprotectin with small intestinal bacterial overgrowth has also been determined in another study [59]. Recent reports suggest that *H. pylori* may colonize in the human gut [69]. Subsequently, the organism may elicit immune responses in the intestine and contributes to intestinal damage as well as pathogenesis of inflammatory bowel disease [69]. Therefore, the elevation of AAT, MPO, and calprotectin in children infected at baseline indicates that infection with *H. pylori* may exerts an adverse impact on intestinal health. Moreover, it also highlights that the effect may persist for a considerable period of time, even after the resolution of infection. We do not know how long the effect may continue and what can be the long-term consequences, particularly on childhood growth. More longitudinal studies are needed to investigate the exact effect of infection with *H. pylori* on the domains of EED and growth of the children.

We observed a negative relationship between *H. pylori* infection and fecal Reg1B concentrations. Reg1B is expressed in human paneth cells and known to have antibacterial effect [70]. Reg1B is also involved in the cellular growth and regeneration [44], and hence, it is supposed to be high in children as they are in the growing stage. A study conducted in Bangladesh also showed that Reg1B concentrations was much higher in children compared to adults [71]. Perhaps, such high amount of Reg1B concentrations with its antimicrobial effect can play a potential role in contributing to the inverse relationship between *H. pylori* and fecal Reg1B concentrations in young children. However, further insight is required to elucidate the interaction between *H. pylori* infection and fecal Reg1B concentrations in children during their early years of life.

Observational studies, conducted in different parts of the world, revealed the adverse impact of *H. pylori* infection on childhood growth and development [72, 73]. Linear growth deficits were much more influenced compared to ponderal growth in *H. pylori* positive

children [72]. In contrast, some studies indicated an absence of association between *H. pylori* infection and nutritional status of the infected children [74–76]. This is what we found as well in our study. There was an insignificant relationship between *H. pylori* infection and the indicators of nutritional status in the children. The children enrolled in this study received nutrition intervention for a period of 90-feeding days. All the fecal biomarker values reduced significantly at the end of intervention indicating improvement of gut health among the children. Probably, the effect of nutrition intervention limited the role of *H. pylori* infection in the children of this cohort. Moreover, we have observed the children only for a shorter period of time, which do not allow postulating the effect of *H. pylori* infection on the growth of the children. The lower prevalence of *H. pylori* infection in the study population can be another cause for such relationship. Timing of growth failure resulting from infection with *H. pylori* should also be taken into consideration. The exact time to potentiate intestinal damage and subsequent growth failure by *H. pylori* is unknown. Additionally, child growth is a complex phenomenon that depends on multiple factors. It is always challenging to control all the confounders to assess growth of young children in community based epidemiological studies. Inability to control all the confounding variables can be another explanation of having insignificant relationship between the infection and growth of children in this study. Therefore, a longitudinal study starting from birth including collection of all the potential confounders' information would be better to elucidate the effect of *H. pylori* infection on childhood growth.

Our study has several shortcomings. First, the participants were malnourished and hailed from a low-resource setting. A comparison group comprising healthy children from improved socio-economic condition would be helpful to understand the impact of *H. pylori* on EED and nutritional status of children. Second, we could not include some of the variables (e.g. birth weight, breast feeding status, pathogen burden, etc.) that were found to be the significant contributors of childhood growth in earlier studies. The strength of the study includes the comprehensive approach to understand the interactions between *H. pylori* infection and consequent gut function during early childhood.

In conclusion, the study results affirmed the acquisition and persistence of *H. pylori* infection during early childhood in the children living in an urban settlement in Dhaka, Bangladesh. Children living in crowded households were more likely to be affected by the infection with *H. pylori*. The indicators of childhood growth were not associated with the infection caused by *H. pylori* in this cohort of children. However, the children with *H. pylori* infection had higher concentrations of AAT, and lower concentrations of Reg1B in their stool samples. This result suggests that *H. pylori* infection may exert an adverse impact on the intestinal health and function, induce loss of gut integrity and result in intestinal inflammation as well as increased gut permeability. This finding would help better understand the etiology and pathophysiology of EED in young children living in resource poor settings. The study results also indicate the importance of *H. pylori* infection in contributing to altered gut function and implicate the importance of further research regarding the role of *H. pylori* infection on EED as well as nutritional status of children during early years of life.

## Supporting information

**S1 Checklist. STROBE checklist.**
(DOC)

**S1 Table. Variables used in this analysis.**
(DOCX)

## Acknowledgments

The authors express thanks to the staff and participants of the BEED study as well as the field and laboratory staffs at icddr,b for their valuable contributions. icddr,b is also grateful to the Government of Bangladesh, Canada, Sweden and the UK for providing unrestricted support.

## Author Contributions

**Conceptualization:** Shah Mohammad Fahim, Ramendra Nath Mazumder, Tahmeed Ahmed.

**Data curation:** Shah Mohammad Fahim, Md. Ashraful Alam.

**Formal analysis:** Shah Mohammad Fahim, Md. Ashraful Alam.

**Funding acquisition:** Tahmeed Ahmed.

**Investigation:** Shah Mohammad Fahim, Subhasish Das, Md. Amran Gazi, Md. Mehedi Hasan, Md. Shabab Hossain, Rashidul Haque.

**Methodology:** Shah Mohammad Fahim, Md. Amran Gazi, Md. Ashraful Alam, Shafiqul Alam Sarker, Ramendra Nath Mazumder, Tahmeed Ahmed.

**Project administration:** Shah Mohammad Fahim, Subhasish Das, Mustafa Mahfuz, Tahmeed Ahmed.

**Resources:** Mustafa Mahfuz, Ramendra Nath Mazumder, Tahmeed Ahmed.

**Supervision:** M Masudur Rahman, Ramendra Nath Mazumder, Tahmeed Ahmed.

**Writing – original draft:** Shah Mohammad Fahim.

**Writing – review & editing:** Subhasish Das, Md. Amran Gazi, M Masudur Rahman, Shafiqul Alam Sarker, Ramendra Nath Mazumder, Tahmeed Ahmed.

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
