## [Decision Letter · Decision Letter 0]

13 Jan 2020

Dear Dr Fahim:

Thank you very much for submitting your manuscript "Helicobacter pylori infection is associated with fecal biomarkers of environmental enteric dysfunction but not with the nutritional status of children living in Bangladesh" (#PNTD-D-19-01719) for review by PLOS Neglected Tropical Diseases. Your manuscript was fully evaluated at the editorial level and by independent peer reviewers. The reviewers appreciated the attention to an important problem, but raised some substantial concerns about the manuscript as it currently stands. These issues must be addressed before we would be willing to consider a revised version of your study. We cannot, of course, promise publication at that time.

We therefore ask you to modify the manuscript according to the review recommendations before we can consider your manuscript for acceptance. Your revisions should address the specific points made by each reviewer with provision of responses/rebuttals/action taken etc. 

When you are ready to resubmit, please be prepared to upload the following:

(1) A letter containing a detailed list of your responses to the review comments and a description of the changes you have made in the manuscript.

(2) Two versions of the manuscript: one with either highlights or tracked changes denoting where the text has been changed (uploaded as a "Revised Article with Changes Highlighted" file); the other a clean version (uploaded as the article file).

(3) If available, a striking still image (a new image if one is available or an existing one from within your manuscript). If your manuscript is accepted for publication, this image may be featured on our website. Images should ideally be high resolution, eye-catching, single panel images; where one is available, please use 'add file' at the time of resubmission and select 'striking image' as the file type. 

Please provide a short caption, including credits, uploaded as a separate "Other" file. If your image is from someone other than yourself, please ensure that the artist has read and agreed to the terms and conditions of the Creative Commons Attribution License at http://journals.plos.org/plosntds/s/content-license (NOTE: we cannot publish copyrighted images). 

(4) If applicable, we encourage you to add a list of accession numbers/ID numbers for genes and proteins mentioned in the text (these should be listed as a paragraph at the end of the manuscript). You can supply accession numbers for any database, so long as the database is publicly accessible and stable. Examples include LocusLink and SwissProt.

(5) To enhance the reproducibility of your results, we recommend that you deposit your laboratory protocols in protocols.io, where a protocol can be assigned its own identifier (DOI) such that it can be cited independently in the future. For instructions see http://journals.plos.org/plosntds/s/submission-guidelines#loc-methods

While revising your submission, please upload your figure files to the Preflight Analysis and Conversion Engine (PACE) digital diagnostic tool, https://pacev2.apexcovantage.com/ PACE helps ensure that figures meet PLOS requirements. To use PACE, you must first register as a user. Then, login and navigate to the UPLOAD tab, where you will find detailed instructions on how to use the tool. If you encounter any issues or have any questions when using PACE, please email us at figures@plos.org.

We hope to receive your revised manuscript by Mar 13 2020 11:59PM. If you anticipate any delay in its return, we ask that you let us know the expected resubmission date by replying to this email.

To submit a revision, go to https://www.editorialmanager.com/pntd/ and log in as an Author. You will see a menu item call Submission Needing Revision. You will find your submission record there. 

Sincerely,

Zulfiqar A. Bhutta, PhD

Associate Editor

Robert Reiner

Deputy Editor

Reviewer's Responses to Questions

**Key Review Criteria Required for Acceptance?**

**Methods**

-Are the objectives of the study clearly articulated with a clear testable hypothesis stated?

-Is the study design appropriate to address the stated objectives?

-Is the population clearly described and appropriate for the hypothesis being tested?

-Is the sample size sufficient to ensure adequate power to address the hypothesis being tested?

-Were correct statistical analysis used to support conclusions?

-Are there concerns about ethical or regulatory requirements being met?

Reviewer #1: This is a carefully designed study with a robust study design undertaken as part of a much larger research programme into environmental enteropathy in Dhaka. The population is carefully described and appropriate although the lack of well nourished controls is acknowledged by the authors as a potential limitation of the study. The sample size is more than adequate and the statistical analysis is thorough and clearly described. Appropriate ethical approvals were obtained for the study protocol

Reviewer #2: Though the objectives are clearly mentioned and testable but the reason for selection of these objectives and study are not clearly mentioned. It is not mentioned why this study is needed and what will be the impact of results of this study. How finding any association or lack of association between H. pylori infection and fecal biomarkers in EED will advance any significant and useful scientific knowledge.

Study design is appropriate but the samples size calculation has not been given. It is not mentioned if recruitment of 319 children was a convenient sample or based on any calculation.

Since, it is secondary analysis of data, details of study population are not needed, though some details have been provided here.

Reviewer #3: A clear hypothesis was not stated: the authors' implied that they sought to investigate the factors associated with H. pylori infection and identify its relationship with the fecal biomarkers of EED. This should be stated clearly.

In general the study was conducted in a straight-forward manner, appropriate methodologies were used to determine H. pylori infection rates and assess biomarkers, and all statistical analysis appears to have been conducted in a thorough and rigorous manner.

No concerns exist about ethical / regulatory requirements. The research protocol of this study (protocol no.: PR-16007) was approved by the Institutional Review Board of the International Center for Diarrheal Disease Research, Bangladesh and written informed consent was obtained from the parents or legal guardians.

The patient population appears to exhibit slightly lower baseline infection rate of H. pylori when compared to reporting by other groups examining infection rates in Bangladesh.

The authors note examples of data appearing to trend in a manner that would support their (likely) preferred conclusion (MPO, Calprotectin, correlation of infection/persistence with stunting). A larger sample size could potentially be one variable that would move these borderline results into the 'significant' category.

**Results**

-Does the analysis presented match the analysis plan?

-Are the results clearly and completely presented?

-Are the figures (Tables, Images) of sufficient quality for clarity?

Reviewer #1: The results follow the analysis plan and are cleary described.Study results confirm the acquisition and persistence of H. pylori infection in early childhood in the children living in an urban settlement in Dhaka, Bangladesh. Children living in crowded households were more likely to be affected by the infection with H. pylori. The children with H. pylori infection had higher concentrations of AAT, and lower concentrations of Reg1B in their stool samples.

Reviewer #2: Though analysis of data is appropriate but table 3 needs to be redesigned. In column of variable, instead of writing H. pylori repeatedly, the markers should have been written. Hence, the table should be redrawn, writing biomarkers in first column and avoiding " Association of Helicobacter pylori infection with ------" in each line.

Reviewer #3: The results are clearly and accurately presented.

**Conclusions**

-Are the conclusions supported by the data presented?

-Are the limitations of analysis clearly described?

-Do the authors discuss how these data can be helpful to advance our understanding of the topic under study?

-Is public health relevance addressed?

Reviewer #1: The authors note that around 20% of their study participants, irrespective of their nutritional status, were infected at enrolment. They conclude that H. pylori infection might have an adverse impact on the intestinal health and function of young children in an urban developing country setting. They also point out that further work is required to elucidate the interaction between H. pylori infection and fecal Reg1B concentrations in children during their early years of life.

Reviewer #2: Though conclusions are supported by the data but are meaningless. The authors have not discussed at all, how this data can be used to advance understanding of the topic. Just finding of presence or absence of relationship between H. pylori infection and fecal biomarkers of EED doesn't mean anything. Similarly, risk factors for H. pylori infections are well known and nothing new has been found in this study.

Reviewer #3: The conclusions of the study and their impact on public health relevance are vague and understated. They should be more clearly presented in the abstract and discussion sections.

The authors do address some of the limitations inherent to their study design.

**Editorial and Data Presentation Modifications?**

Reviewer #1: This is a well-written paper and I have no suggestions for editorial changes.

Reviewer #2: None.

Reviewer #3: (No Response)

**Summary and General Comments**

Reviewer #1: This study provides a comprehensive approach to understand the interactions between H. pylori infection and consequent gut function during early childhood.The adverse impact of H.pylori infection on childhood growth and development well reported but this study showed insignificant relationship between H. pylori infection and the indicators of nutritional status in the children. The authors comment on the paucity of research investigating the relationship between H. pylori infection and fecal concentrations of AAT. Authors may be interested in the finding from the reviewer in The Gambia (Sullivan PB, Thomas JE, Eastham EJ, Lunn PG, Neale G. Helicobacter pylori and protein losing enteropathy. Archives of disease in childhood. 1990;65(3):332-3.) which found no association between H.pylori and fecal AAT levels in malnourished children with persistent diarrhoea.

Reviewer #2: The manuscript is based on a secondary analysis of data obtained during study on environmental enteric dysfunction, which is already a poorly understood condition. Writing of this manuscript is a share wastage of time, effort and money. This newly described condition, previously known by various names such as tropical enteropathy, etc is not yet an established single disease, nor its diagnostic criteria are well defined. Various blood and stool markers are being investigated, some of those are used in this study and manuscript. In addition, H. pylori has been given an undue importance, role of which has yet to be defined in causation of the EED. Looking for a relationship between H. pylori and fecal markers of EED without any sound hypothesis or reason is not going to lead anywhere. H. pylori infects gastric mucosa, whereas EED is a disease of small intestine. Hence, a strong reason needs to be given to look at the relationship of H. pylori with fecal markers of EED, which has not been given. Even if some relationship is found between H. pylori infection and few markers of EED, it doesn’t explain any cause or effect. Hence, the manuscript is not fit for publication and may be rejected. 

In addition to above comments, there are several other points which need to be taken into consideration and are discussed below. 

1. Stunted and at risk for stunting: I am not sure why the term "at risk for stunting" has been used in this manuscript. Why the authors have labelled children with normal height as "at risk for stunting"? Specially when already a nutrition intervention program is going on. Thus, the children should be labeled as "stunted" and "not stunted".

2. Regarding H. pylori infection, it is mentioned that one-fifth of the participants had H. pylori infection at both the time points, with 13.8% overall persistence. These children should have been divided into three groups, a) Those who had infection at recruitment but got cleared by the end of study. b) those children who remained infected and c) who acquired new infection during the study.

These three groups should be analyzed separately and compared. This probably could show if children with abnormal fecal biomarkers are prone to H. pylori infection or H. pylori infection led to abnormal fecal biomarkers and EED.

3. To look at any association between H. pylori infection and fecal biomarkers, normal children without EED should have been included in the study. Only then, one would be able to determine if H. pylori infection has any relationship with these fecal biomarkers. If any such association is found in normal children i.e. if normal children without EED but infected with H. pylori are found to have these fecal biomarkers in stool, it may be inferred that H. pylori may have a role in causation of EED. If not so then EED might predispose children to acquire H. pylori infection. Presence or absence of relationship between H. pylori and fecal biomarkers without looking at the pathogenesis or cause and effect doesn't mean anything, particularly when H. pylori group is a heterogenous group.

4. Tables should be drawn in standard way without any grid line.

5. Table 3 is poorly drawn. In column of variable, instead of writing H. pylori repeatedly, the markers should have been written. Hence, the table should be redrawn, writing biomarkers in first column and avoiding " Association of Helicobacter pylori infection with ------" in each line.

Reviewer #3: The authors sought to investigate the factors associated with H. pylori infection and identify its relationship with the fecal biomarkers of EED including Myeloperoxidase (MPO), Neopterin (NEO), Calprotectin, Reg1B and Alpha-1 antitrypsin (AAT), and nutritional status of the children in their patient cohort in Bangladesh. The study found a positive association with AAT, an inverse relationship with Reg1B, and no association with the other biomarkers examined, nutritional state and/or stunting in children. The authors state that their findings affirmed the acquisition and persistence of H. pylori infection, which they claimed might account for altered gut function in the early years of life.

The overall concept defining the study is interesting and novel: a correlative study investigating H. pylori infections with the potential to exacerbate EED has not been carried out previously and given the high prevalence of both in this patient population, investigation into how each condition influences the other and patient outcomes is of high importance to the field.

In general the study was conducted in a straight-forward manner, appropriate methodologies were used to determine H. pylori infection rates and assess biomarkers, and all statistical analysis appears to have been conducted in a thorough and rigorous manner. The manuscript is well organized and written clearly enough to be accessible to non-specialists. The only major issue with the manuscript in its current form is that the conclusions that the authors draw from the data are somewhat hard to interpret. This may be due to the relative ‘mixed bag’ of their reported results, however, the authors need to try and address these potential discrepancies in the context of their hypothetical model. In this light, the project’s overall goals and overarching hypothesis need to be more clearly stated in the abstract, discussion, and conclusion sections. I have provided some ideas on how the authors might wish to address this below:

Major comments:

The major limitation of this study is its lack of a clearly-defined, testable research objective, and the consequent weakness of the authors’ overall conclusions. ‘Hypothesis’ was not mentioned once in the manuscript, but if one was to be inferred from the author’s introduction, it would be that H. pylori infections contribute to and/or exacerbate Environmental Enteric Dysfunction (EED) and subsequent growth failure in children. In their conclusions, rather than state whether this hypothesis was supported (or not) by the data generated in this study, the author’s hedge and conclude what is already well-known to the field; that the acquisition and persistence of H. pylori infection occurs early in life. If this is the only definitive conclusion of this study, then it is not of sufficient novelty for publication in PLOS Neglected Tropical Diseases … because we already know this. The novelty comes from whether or not H. pylori infection exacerbates EED and the authors need to do 1 of 3 things to address this potential hypothesis:

1) State that infection does correlate with an exacerbated EED status based on the AAT / Reg1B data and offer a convincing explanation (preferably backed up with either experimentation or the citation of the experimentation of others) as to why AAT (a readout of enteric protein loss / intestinal permeability) and Reg1B (tissue injury/repair) might be better indicators of EED status in children than MPO, NEO, and Calprotectin (which offer readouts of inflammation)

2) State that exacerbated EED status should result in elevated levels for all biomarkers examined (AAT, MPO, NEO, and Calprotectin) and thus conclude that H. pylori infection does not appear to exacerbate EED in this patient cohort

3) State that the data was inconclusive and suggest reasons for this (potentially due to inadequate sampling size, robustness of the assay, biomarkers of EED failing to capture a perfect picture of disease abundance in general, etc.)

Given that EED has been previously shown to affect MPO, NEO, Calprotectin, child growth/development/nutritional status and that none of these appear to be different between Hp infected and non-infected individuals in this study, the data would appear to favor #2.

It is curious that MPO, Calprotectin, and NEO (readouts for intestinal inflammation) were not significantly different between infected and non-infected individuals, and yet both AAT (associated with protein loss / intestinal permeability) and Reg1B (associated with epithelial tissue injury and subsequent repair) were significantly different. Regardless of their eventual decision on the overall conclusions of this study, the authors should present an explanation for why epithelial tissue injury and intestinal protein loss (presumably from an active infection) might not result in subsequent inflammation (which is generally highly abundant during Hp infections, although presumably less so in the small intestine).

Line 42-43: Consider modification to: “Infection was not associated with nutritional status of the children, nor was it associated with indicators of childhood growth.”

Lines 68-89: Isenbarger et al (1998) found that the incidence of diarrhea was not significantly different between children by H. pylori serostatus and concluded that H. pylori infection was not associated with an increased risk of diarrheal disease. Quinonez et al (1999) determined that Helicobacter pylori appeared to have no effect on the nutritional status of the studied children and that differences detected were small and likely due to sociodemographic factors. Vilchis et al (2009) was the only one of the three studies cited to suggest that H. pylori infection has a negative effect on the growth of children. A better citation would be Franceschi et al 2014, World Journal of Gastroenterology (and the articles cited therein) as they summarize a number of studies that have found positive associations with H. pylori incidence and malnutrition as well as explore mechanistic aspects of H. pylori-caused malnutrition. Overall (as the authors imply in lines 76-77 and later on in the discussion section) the data on the topic reported to date are decidedly a mixed bag. The introduction should also make this clear.

Figure 1: As the authors note, the number of H. pylori infected individuals in the author’s dataset appears to be on the low side for Bangladesh, with only 20% of the participants testing positive for H. pylori. Rates of infection of up to 60% have been reported in the population at large (Habib 2016) and in infants aged 1-3 months (Mahalanabis 1996, who also reported finding no association with H. pylori infection and nutritional state). The author’s should comment on how this might affect the interpretation of their results.

Line 215-216: " ... all the fecal biomarker values were reduced significantly after the 90-day nutrition intervention (p-value <0.05)." This would seem to be a rather significant finding: do the authors take this as evidence of successfully combating EED with nutritional intervention? If so, they should state as much.

Line 259-260: This data is in line with previous reports (most recently Aitila 2019) showing higher prevalence in crowded households.

PLOS authors have the option to publish the peer review history of their article (what does this mean?). If published, this will include your full peer review and any attached files.

Reviewer #1: Yes: Peter B Sullivan

Reviewer #2: No

Reviewer #3: No

---

## [Editor Report · Decision Letter 1]

23 Mar 2020

Dear Dr Fahim,

We are pleased to inform you that your manuscript 'Helicobacter pylori infection is associated with fecal biomarkers of environmental enteric dysfunction but not with the nutritional status of children living in Bangladesh' has been provisionally accepted for publication in PLOS Neglected Tropical Diseases.

Best regards,

Zulfiqar A. Bhutta, PhD

Associate Editor

Robert Reiner

Deputy Editor

---

## [Editor Report · Acceptance letter]

15 Apr 2020

Dear Dr Fahim,

We are delighted to inform you that your manuscript, "Helicobacter pylori infection is associated with fecal biomarkers of environmental enteric dysfunction but not with the nutritional status of children living in Bangladesh," has been formally accepted for publication in PLOS Neglected Tropical Diseases.

Best regards,

Serap Aksoy

Editor-in-Chief

Shaden Kamhawi

Editor-in-Chief
